# Phylogenetic Analysis and Codon Usage Bias Reveal the Base of Feline and Canine Chaphamaparvovirus for Cross-Species Transmission

**DOI:** 10.3390/ani13162617

**Published:** 2023-08-14

**Authors:** Xu Guo, Yingying Zhang, Yang Pan, Kankan Yang, Xinxin Tong, Yong Wang

**Affiliations:** 1College of Animal Science and Technology, Anhui Agricultural University, Hefei 230036, China; 2Institute of Infectious Diseases, Shenzhen Bay Laboratory, Shenzhen 518000, China

**Keywords:** canine, feline, *Chaphamaparvovirus*, codon usage, phylogenetic, cross-species transmission

## Abstract

**Simple Summary:**

*Chaphamaparvovirus*, a significant genus of the *Hamaparvovirinae* subfamily of the *Parvoviridae* family, can infect both dogs and cats. Given the evidence of cross-species transmission observed in canine and feline parvoviruses, it is pertinent to investigate the potential for similar cross-species transmission with *Chaphamaparvovirus*. This study aimed to investigate the basis for the ability of FeChPV and CaChPV to undergo cross-species transmission by evaluating phylogenetic analysis and codon usage analysis. The phylogenetic analysis revealed a close relationship between canine and feline chaphamaparvoviruses, and their strong adaptation to dogs and high similarity in codon usage patterns suggests the possibility of unidirectional or bidirectional cross-species transmission between dogs and cats. These findings underscore the importance of monitoring and examining the risks associated with cross-species transmission of canine and feline chaphamaparvoviruses.

**Abstract:**

Chaphamaparvoviruses (ChPVs) are ancient viruses that have been detected in a variety of hosts. In this study, through a phylogenetic analysis and the adaptability of ChPV to multiple hosts, we evaluated the basis for the ability of feline (FeChPV) and canine ChPV (CaChPV) for cross-species transmission. Phylogenetic analysis showed that FeChPV and CaChPV were closely related. Notably, two strains of ChPVs isolated from domestic cats and two from dogs clustered together with CaChPVs and FeChPVs, respectively, suggesting that the stringent boundaries between canine and feline ChPV may be broken. Further analysis revealed that CaChPV and FeChPV were more adapted to dogs than to cats. Mutation analysis identified several shared mutations in cross-species-transmissible strains. Furthermore, the VP structures of FeChPV and CaChPV exhibited a high degree of similarity across both cross-species-transmissible and non-cross-species-transmissible strains. However, it is crucial to note that these results are largely computational, and limitations exist in terms of the number and diversity of samples analyzed; the capacity for cross-species transmission should be approached with caution and elucidated in further studies.

## 1. Introduction

The *Parvoviridae* family is divided into three subfamilies as follows: *Densovirinae*, *Hamaparvovirinae*, and *Parvovirinae* [1]. As the singular member of the *Hamaparvovirinae* subfamily, viruses within the genus *Chaphamaparvovirus* can infect a wide range of hosts, including vertebrates and invertebrates [2]. ChPV can infect various animals, including bats [3], rodents [4], birds/poultry [5,6,7,8], pigs [9], dogs [10], domestic cats [11], Tasmanian devils [12], wild mammals [13,14], and humans [15]. Canine ChPV (CaChPV) and feline ChPV (FeChPV) are classified as species *Carnivore chaphamaparvovirus 1* and *Carnivore chaphamaparvovirus 2*, respectively [11,16]. The complete genome of FeChPV/CaChPV is approximately 3400 bp in length and contains two major open reading frames: nonstructural protein (NS) and virion protein (VP).

In 2017, a novel parvovirus, termed cachavirus, was detected in the feces of dogs during an outbreak of diarrhea of an unexplained origin [10]. It was subsequently detected in Canada, China, and Italy [16,17,18]. Similarly, FeChPV was first detected in Canada during an outbreak of unknown origin in an animal shelter [19]. Both CaChPV and FeChPV are supposedly associated with gastrointestinal symptoms and are more frequently detected in animals with diarrhea than in healthy animals. Furthermore, FeChPV has been detected in cats with respiratory diseases and coinfections with other common viruses [20].

Previous studies have reported on the cross-host transmission of parvoviruses. Canine parvovirus (CPV) 2a/2b/2c infects cats and causes symptoms similar to those caused by feline parvovirus (FPV) infection in domestic cats [21,22]. Mink parvovirus is a variant resulting from the adaptation of the FPV to mink. Interestingly, in China, CaChPV has been detected in cats, while FeChPV has been found in dogs, strongly suggesting host tropism plasticity [17,23].

In the present study, we focused on the potential of FeChPV and CaChPV for cross-species transmission. A phylogenetic analysis and codon usage analysis revealed the potential host range diversity of the carnivore ChPV and provided additional information to support the understanding of ChPV.

## 2. Materials and Methods

### 2.1. Clinical Samples and Viral Genome Sequence Collection

From July 2018 to October 2020, 58 fecal samples from domestic cats were collected using anal swabs from different animal hospitals in Anhui province. Thirty-two swabs were obtained from cats with diarrhea, whereas the remaining samples were obtained from healthy cats without diarrhea symptoms. Virus extraction, polymerase chain reaction, and sequencing were performed as previously described [24]. This study was approved by the Institutional Animal Care and Use Committee of Anhui Agricultural University (SYDW-P20200311059).

We collected reference sequences of FeChPV, CaChPV, porcine parvovirus 7, poultry/birds ChPV, mouse kidney parvovirus (*Mus musculus*), bat ChPV, hedgehog (*Erinaceus amurensis*) ChPV, pangolin ChPV, red fox (*Vulpes vulpes)* ChPV, American black bear (*Ursus americanus)* ChPV, Tasmanian devil (*Sarcophilus harrisii*) ChPV, human (*Homo sapiens)* ChPV, and primate (both *Cebus capucinus imitator* and *Macaca fascicularis*) ChPV from the National Center for Biotechnology Information (NCBI, https://www.ncbi.nlm.nih.gov/, accessed on 13 December 2022) (Appendix A). 

### 2.2. Similarity, Genetic Distance, and Phylogenetic Analysis

Similarity analysis of the sequences of FeChPV and CaChPV was performed using Megalign software (DNASTAR, Madison, WI, USA). MEGA X 10.1.7 was used for genetic distance analysis using the *p*-distance model with 1000 bootstraps.

The sequences were aligned using multiple alignment with fast Fourier transform [25] and trimmed using the TrimAl softwareV1.2 [26]. The optimal substitution model was selected using ModelFinder [27]. We inferred maximum-likelihood phylogenies using IQ-TREE with ultrafast bootstrap analysis [28]. Phylogenetic trees were visualized using FigTree version 1.4.3 (http://tree.bio.ed.ac.uk/software/figtree, accessed on 25 December 2022) and Interactive Tree Of Life [29].

### 2.3. Codon Usage Analysis

We analyzed the nucleotide compositions of NS and VP in FeChPV and CaChPV. The basic nucleotide composition (A%, T%, C%, and G%), nucleotides at the third position of synonymous codons (A3s%, T3s%, C3s%, and G3s%), and G/C content at the third synonymous codon position (GC3s) were calculated using CAIcal SERVER [30].

The effective number of codons (ENC) was also calculated using the CAIcal server. ENC values range from 20 to 61, where an ENC less than 35 indicates a codon usage bias (CUB); a smaller value denotes stronger bias. The ENC formulae were based on previous studies [31].

### 2.4. Relative Synonymous Codon Usage Analysis 

We compared the codon usage patterns of FeChPV and CaChPV with those of the host. We analyzed the relative synonymous codon usage (RSCU) of the FeChPV and CaChPV genomes. The RSCU value indicates the ratio of observed codon occurrence to random occurrence. RSCU helps to understand the preferential use of synonymous codons. A synonymous codon with a higher frequency of occurrence has an RSUC > 1, whereas that with a lower frequency has an RSCU < 1. An RSCU > 1.6 denotes an overrepresented synonymous codon, whereas one of <0.6 denotes an underrepresented synonymous codon [32]. The preferred codon was defined as the one most used for an amino acid (one with the highest RSCU value). The RSCU was calculated using CAIcal SERVER [30]. All coding sequences of each strain were included in the analysis. The RSCU of the host was calculated using the following equation:$${RSCU}_{ij}=\frac{X_{ij}}{\sum_{j=1}^{ni}X_{ij}}ni$$
where *RSCU_ij_* is the value of the *i*-th synonymous codon of the *j*-th amino acid, *X_ij_* is the observed number of the *i*-th codon of the *j*-th amino acid, and “*ni*” denotes the number of synonymous codons that encode the *j*-th amino acid.

We defined the dissimilarity in synonymous codon usage between viruses and hosts using Euclidean distances (*Dp*):$$D_{i}=\sqrt{\sum_{j=1}^{ni}{\left(y_{ij}-x_{ij}\right)}^{2}}$$
where *ni* is the number of synonymous codons of amino acid *i*, *y_ij_* is the fraction of codon *j* among synonymous codons of amino acid *i* in the viral gene, and *x_ij_* is the supply of tRNA represented by the fraction of codon *j* among the synonymous codons in the host transcriptome. *Dp* was defined as the weighted set average of the equivalent weights of the *D_i_* value of 18 amino acids. Higher *Dp* values suggest greater dissimilarity, while lower values indicate less dissimilarity [33].

### 2.5. Codon Usage Pattern Difference Analysis

Relative codon deoptimization index (RCDI) analysis was performed using the RCDI server [34]. An RCDI equal to 1 indicates that the virus displayed a pattern of codon usage adapted to the host. Conversely, a value >1 indicates low adaptation [35]. A higher RCDI value indicates a greater variance from the codon usage pattern of the host. The host codon usage was collected from the codon-and-codon pair usage tables (CoCoPUTs) [36].

We used the similarity index [SiD or D (A, B)] to estimate the influence of the host codon usage patterns on virus formation. SiD values ranged from 0 to 1, with higher values indicating a stronger host influence on viral codon usage and less host adaptation. Lower values indicated the converse, i.e., higher adaptability of the host. We used the following formula:$$R\left(A,B\right)=\frac{\sum_{i=1}^{59}a_{i}b_{i}}{\sqrt{\sum_{i=1}^{59}{b_{i}}^{2}\sum_{i=1}^{59}{a_{i}}^{2}}}$$
where *a_i_* denotes the RSCU value of the 59 synonymous codons of the virus coding sequence, and *b_i_* denotes the RSCU value of the identical codon in the host [35].

### 2.6. Host Adaptability Analysis

The codon adaptation index (CAI) was estimated using the CAI calculation of the CAIcal server [30]. The host codon tables of *Canis lupus familiaris*, *Felis catus*, *Sarcophilus harrisii*, *Chiroptera*, *Manis*, *Mus musculus*, *Ursus americanus*, *Macaca fascicularis*, *Macaca mulatta*, and *Homo sapiens* were collected from the codon-and-codon pair usage tables (CoCoPUTs) [36]. For *Canis lupus familiaris* and *Felis catus*, we used normalized CAI (nCAI) to further correct the CAI values [37]. Normalized CAI was defined as the quotient between the CAI and its expected CAI (eCAI). eCAI was estimated using the CAI calculation of the CAIcal server [30]. An nCAI less than 1 is considered to denote differences in CAI values due to nucleotide composition, while values of nCAI closer to 1 or greater than 1 can be interpreted as evidence of adaptation to the host codon usage pattern [37]. 

### 2.7. Parity Rule 2, ENC Plot, and Neutrality Analysis

We performed a parity rule 2 (PR2) analysis to evaluate whether the influence of selection pressure on codon usage during evolution was consistent with mutations. A3/(A3 + T3) was the abscissa, G3/(G3 + C3) was the ordinate, and the coordinate axis at 0.5 was the origin. The effects of selection pressure and mutations were considered inconsistent in cases with dots clustered around the origin and consistent in other cases [38].

ENC plots use ENC as the ordinate and GC3s as the abscissa. Cases with dots clustered on/above and under the curve indicate that the CUB was greatly affected by mutations and other factors (e.g., selection pressure) [39].

The equation for calculating the expected ENC value was as follows:$${ENC}_{expected}=\frac{29}{x^{2}-{\left(1-x\right)}^{2}},$$
where *x* is the frequency of GC at the third position of synonymous codons.

### 2.8. Comparative Analysis of Mutations, Immune Epitopes, and Structures

The VP protein sequences of FeChPV and CaChPV were created separately as separate datasets. Protein sequence comparisons were performed by MEGA X [40]. Mutation analysis was performed using the ESPript 3.0 (https://espript.ibcp.fr, accessed on 23 April 2023) [41]. B-cell immune epitopes were predicted using SVPPriT, and scores greater than 0.7 were adopted [42]. The structures of the Vp genes of CaChPV/Cat/MN928790.1, CaChPV/Cat/MN928791.1, CaChPV/Dog/MT123284.1, FeChPV/Dog/OQ162042.1, FeChPV/Dog/OQ162043.1, and FeChPV/Cat/MN396757.1 were predicted using AlphaFold2 [43]. We used PyMOL Molecular Graphics System (Version 2.0 Schrödinger, LLC., New York, NY, USA) for visualization and comparative analysis of the protein structures. The root-mean-square deviation (RMSD) was used to measure the size of protein structural differences, where an RMSD of 1 Å was considered a cut-off for different structures [44]. 

### 2.9. Statistical Analysis

The *Shapiro–Wilk* test was conducted as the normality test. We performed the *Mann–Whitney U* test and one-way analysis of variance to analyze the significance of the Gaussian distribution data and non-conforming data, respectively. Graphpad Prism v 9.3.0 was used for the statistical analysis and data visualization.

## 3. Results

### 3.1. Clinical Samples

A total of 58 stool samples were examined, and four were positive for FeChPV. The positivity rate was 6.9% (4/58). All four samples were collected from cats diagnosed with gastroenteritis, with a positivity rate of 12.5% (4/32). FeChPV was not detected in asymptomatic cats. We amplified copies of the entire genome from the positive sample and sent it to a Sangon Company (Chuzhou, China) for sequencing. The complete genome sequences were uploaded to GenBank (Accession numbers: MT708230.1/HF1, MT708231.1/HF2, MZ031965/AH-03, and MZ031966.1/04).

### 3.2. Similarity and Genetic Distance

To evaluate the similarity and genetic distance between FeChPVs and CaChPVs at both the nucleotide and the amino-acid levels, we implemented genome-wide and gene-specific similarity and genetic distance analyses. Over 95% similarity was preserved within the FeChPV and CaChPV groups at the genomic level, as well as within the NS and VP genes. Conversely, less than 75% similarity was observed between the FeChPV and CaChPV groups. The genetic distance analysis corroborated the results from the gene-specific similarity analysis. To conclude, the intra-group and inter-group genetic distances of FeChPV and CaChPV demonstrated substantial differences of at least one order of magnitude (Appendix A).

We analyzed the structural similarity of VPs between cross-species transmitted and non-cross-species transmitted strains. Our findings demonstrated that both cross-species transmitted and non-cross-species transmitted strains displayed highly similar VP protein structures (RMSD < 1) for FeChPV and CaChPV, suggesting their analogous receptor-binding configurations (Figure 1). Nonetheless, the presence of mutations may result in variations in binding affinity. Owing to the absence of receptor information for ChPV, we could not determine the consistency between individual virus receptor-binding ability and binding sites. 

### 3.3. Phylogenetic Analysis

Phylogenetic trees were established on the basis of complete genomes, as well as NS and VP. In the genome-wide tree (Figure 2), FeChPV and CaChPV clustered on the same branch and in the same lineage as ChPV isolated from rodents (*Mus musculus*), *Sarcophilus harrisii*, *Ursus americanus*, primates (*Cebus capucinus imitator* and *Macaca fascicularis*), hedgehog (*Erinaceus amurensis*), pangolins, and bats, thus indicating a close relationship linking CaChPV, FeChPV, and ChPV isolated from these hosts. Further analysis of the phylogenetic trees of NS and VP revealed that FeChPV and CaChPV belonged to two different branches. However, two strains of ChPV isolated from domestic cats clustered together with CaChPVs, and two stains isolated from dogs clustered together with FeChPVs, thereby suggesting potential host spillover events exist for FeChPV and CaChPV (Figure 3).

### 3.4. Codon Usage Pattern Difference Analysis

To more accurately quantify the disparity in the codon usage patterns between FeChPV and CaChPV and their hosts, we conducted an analysis utilizing the RCDI and SiD. According to the RCDI (Figure 4), FeChPV and CaChPV had lower RCDI values for dogs than for cats, suggesting that FeChPV and CaChPV may have higher potential adaptation for dogs compared to cats. The codon usage patterns of CaChPV were more similar to those in dogs and cats, but not significantly different from those of FeChPV (*p* > 0.05). The SiD results were similar to those of RCDI (Figure 4). 

### 3.5. Host Adaptability Analysis

Variations in CUB are associated with adaptations to different hosts. We performed an adaptive analysis of dogs and cats (Appendix A). Results of the CAI (Table 1) indicated that the ability of the FeChPV and CaChPV genes to be highly expressed in both canine and feline host cells. The results for eCAI and nCAI were also identical. These results are consistent with the results for RCDI and SiD. 

### 3.6. CUB and RSCU Analysis

We observed deviations in codon usage for the same virus from different hosts. Therefore, we analyzed codon usage preferences and their relationships with hosts. In the NS, neither FeChPV nor CaChPV displayed a strong CUB, with mean ENC values of 43.123 ± 0.461 and 45.047 ± 0.613, respectively. However, a stronger CUB was identified in the VP of the FeChPV group (35.081 ± 0.417) (Table 2 and Appendix A). 

CUB leads to a difference in usage of synonymous codons. We, therefore, analyzed the difference between the RSCU of FeChPV and CaChPV and the RSCU of the host to assess differences in synonymous codon preference. The RSCU values of CaChPV and FeChPV were much less different from those of dogs, indicating that the codon use preference of CaChPV and FeChPV was more similar to that of dogs (Figure 5 and Appendix A).

### 3.7. Parity Rule 2, ENC Plot, and Neutrality Analysis

The PR2 analysis demonstrated that neither the NS nor the VP of FeChPV and CaChPV was clustered at the origin, thereby indicating that both selection and mutation affected the CUB of these viruses in a manner that was consistent throughout the evolution (Figure 6a). The ENC plot depicted that all dots were distributed below the expected curve, which indicated that the selection pressure played a dominant role in these genes (Figure 6b). Consistent with these findings, the slopes of the regression lines were −0.1755, 0.08417, 0.01133, and 0.05754, respectively, in the neutrality analysis (Figure 6c), all of which were distant from 1. This aspect demonstrated the primary role of selection pressure in the formation of codon usage preferences for the FeChPV and CaChPV genes. The contributions of natural selection were 82.45% (NS of FeChPV), 91.58% (VP of FeChPV), 98.87% (NS of CaChPV), and 94.25% (VP of CaChPV).

### 3.8. Comparative Analysis of Mutations, and Immune Epitopes

The obtained results demonstrate the influence of mutation and selection pressure on the evolution of both FeChPV and CaChPV. Therefore, we focused on the mutation characteristics of strains known to have been transmitted between species (CaChPV: MN928790.1, MN928791.1; FeChPV: OQ162042.1, OQ162043.1). Among the two FeChPV strains undergoing cross-species transmission, four shared mutations were identified: G112R, L174P, S197T, and S325R. The S197 mutation in both FeChPV strains coincided with the T observed in CaChPV. Furthermore, S206 and S445 in FeChPV-OQ162042.1 mutated to the same A and N as CaChPV, while D402 in CaChPV-MN928790.1 mutated to the same N as in FeChPV. Additionally, the F131S mutation in CaChPV-MN928790.1 was also detected in FeChPV-OQ162042.1 (F151S) (Table 3 and Appendix A).

To examine whether mutations contribute to changes in B-cell immune epitopes, the immune epitopes of both FeChPV and CaChPV were analyzed. The findings revealed no substantial differences in immune epitopes between cross-host transmitted and non-cross-host transmitted strains in either FeChPV or CaChPV. A comparison between FeChPV and CaChPV identified a conserved amino-acid sequence (SIAYKEGMFK) present in the immune epitopes of both viruses (Table 4). 

## 4. Discussion

*Chaphamaparvovirus* is a recently characterized genus within the *Parvoviridae* family, exhibiting a broad host reservoir, encompassing both vertebrate and invertebrate species. ChPV is supposedly an ancient virus existing in animal hosts for millions of years [45]. Further, transmission may have occurred between distantly related host species [45]. In this study, we detected FeChPV in fecal samples from domestic cats; all positive samples were obtained from domestic cats with diarrhea, thus suggesting that FeChPV is associated with diarrhea, similar to CaChPV [39]. ChPV is often associated with feces, thus strongly suggesting an association with intestinal disease [45]; however, it supposedly has the potential to cause other diseases, such as respiratory disease [11], hepatitis [46], encephalitis [13], and chronic interstitial nephropathy [47], in other species. FeChPV was identified in cats exhibiting respiratory symptoms; however, the samples gathered in this study did not include the respiratory tract. As a result, the potential association between the ChPV and respiratory diseases cannot be conclusively assessed, warranting further exploration and study. 

We evaluated the between-group and within-group differences in FeChPV and CaChPV by genetic distance and similarity analyses. Results demonstrated higher between-group similarity and genetic distance than within-group measurements for both individual genes and genomes. The genetic distance displayed a difference of at least one order of magnitude. Surely, this difference may derive from species differences. These findings are reminiscent of the intertransmissibility observed between canine and feline parvoviruses (CPV and FPV), despite genomic differences [21,22]. However, the differences between FeChPV and CaChPV appear to be more significant than those between CPV and FPV. Protein structure analysis revealed RMSD values less than 1, indicating high structural similarity in their VP proteins, which might ease transmission between species.

Previous research identified CaChPV in domestic cats and FeChPV in dogs [17,23], as also substantiated by our phylogenetic findings. FeChPV and CaChPV exhibit intermixing relationships. In addition, FeChPV and CaChPV are closely related to ChPV isolated from rodents (*Mus musculus*), *Sarcophilus harrisii*, *Ursus americanus*, primate (*Cebus capucinus imitator* and *Macaca fascicularis*), hedgehog (*Erinaceus amurensis*), pangolins, and bats. This could be a potential red flag. Wildlife species such as pangolins, bats, and hedgehogs are recognized as natural reservoirs for numerous viruses, despite frequent host spillover events [48,49]. In addition, the close relationship between dogs and cats and humans exacerbates our concerns about the threat that ChPV poses to humans. Given the close association between humans and companion animals such as dogs and cats, a potential for unidirectional or bidirectional cross-host viral transmission exists [50,51], thus necessitating further epidemiological studies to better understand the relationship linking FeChPV, CaChPV, and ChPV strains in humans and wildlife.

In order to explore the codon usage underlying the cross-species transmission that occurs in FeChPV or CaChPV, we analyzed the CAI, RSCU, and codon usage patterns of these two viruses. The CAI value is used to assess the expression of an exogenous gene within the cell. A high CAI value represents high levels of gene expression and a closer match to the host’s codon usage preferences, which may replicate more efficiently [30]. Our CAI analysis suggested that FeChPV and CaChPV have highly adapted to canine hosts, potentially indicating greater expression in dogs as compared to cats. The results of the RCDI, SiD, and D*p* analyses suggested a stronger adaptation to canine codon usage patterns over feline ones. Usually, gene expression requires the aid of transfer RNA (tRNA), and the abundance of tRNA corresponding to the codon used in the host gene is higher within the host cell. The differential codon use pattern would limit the rate of viral gene replication because of the use of a low abundance of tRNA. Generally, for exogenous genes, a closer pattern of use to the host codon indicates more efficient gene expression and greater harm to the host because it inhibits the expression of the host gene [33,52]. Consequently, these analyses suggested that FeChPV and CaChPV display stronger adaptability to dogs, with a potential for effective expression within host cells. Given the proven occurrence of cross-species transmission, these traits potentially form a key foundation for FeChPV and CaChPV cross-species transmission. Notably, the detection of ChPV in the intestine may be a result of food residues caused by a predatory relationship, suggested by the high correlation between ChPV and feces [45]. In one study, ichthyic ChPV was detected in tilapia-fed crocodiles [53]. However, the risk of this phenomenon in cats and dogs is low, as these animals do not have a predatory relationship. Therefore, the potential for transmission of ChPV between dogs and cats is high, and the risk of transmission to other hosts should be monitored. 

Despite the cross-host transmission having been tentatively demonstrated, detection of FeChPV in dogs and CaChPV in cats is rare. This might be attributed to overlooked cross-species transmission properties or a lack of detection attempts. Another potential explanation lies in the fact that, although FeChPV and CaChPV are highly adaptive to canines, their entry into host cells necessitates specific receptors. For instance, CPV requires the transferrin receptor, and previous studies on CPV and FPV suggested that mutations at certain specific loci may endow the viruses with the ability to bind heterologous host receptors [54]. Accordingly, we focused on mutations in four cross-species transmission strains. In the cross-species propagating strains of FeChPV, four common mutation sites were identified, of which S197T is a site of interest because of its mutation from S, which is present in FeChPV, to T, which is present in CaChPV, whereas, in CaChPV, only one site was identified. Mutations at a single locus or the accumulation of mutations at certain loci may lead to the acquisition of the ability to infect other hosts. These mutations do not appear in corresponding B-cell immune epitopes, thus exerting a minimal impact on B-cell immunity. Partial immune epitope overlap between CaChPV and FeChPV suggests potential cross-immunity, although experimental validation is required.

We also attempted to analyze the CAI, RCID, and Sid values of FeChPV and CaChPV for other animals, and the results showed similar characteristics in both dogs and cats. Additionally, the characteristics indicated an especially high degree of adaptation to marsupials. Given that ChPV is an ancient virus, the high degree of adaptation to ancestral marsupials is intriguing. This was further demonstrated by FeChPV and CaChPV being more closely related to marsupial ChPV in the phylogenetic tree (Figure 2). Further exploration is needed to determine whether transmission occurred from early-diverging marsupials to other animal species. Since FeChPV and CaChPV have not been tested in these animals, we are not sure whether ChPV can also be subject to the same wild-type cross-host events as some other viruses. 

The gene mutation pressure, natural selection pressure, secondary protein structure, and external environment are the chief factors contributing to codon bias [24]. Therefore, we conducted PR2, ENC plot, and neutrality analyses to evaluate the key dynamics influencing the CUB. Although both mutation and selection pressure play a role in the evolution of codon usage, the selection pressure was the primary influence, similar to CPV, FPV, and feline bocavirus [24,55]. CPV-2 continually generates new subtypes, such as CPV-2C, which have the potential for cross-species transmission [21]. Selection pressure in virus evolution refers to the various factors and conditions (such as host immune response, antiviral drug and vaccine use, and environmental variables) that drive changes in viral populations over time. These pressures can influence the survival and replication of viral strains, with some variants becoming more successful due to their specific adaptations. The selection pressure drives the process of natural selection, which can result in the emergence of new viral strains with improved fitness in a given environment [56,57,58,59]. Therefore, the selective pressure on CaChPV and FeChPV warrants attention, and further monitoring of the virus should be intensified to assess its evolutionary status.

## 5. Conclusions

In this study, we elucidated the basis for the ability of FeChPV and CaChPV to undergo cross-species transmission through multiple analyses: high adaptability to hosts, potential mutations, and the driving force of potential selective pressures. Significantly, while our study provides a substantial foundation for understanding the genetic basis of FeChPV and CaChPV adaptability and cross-species transmission, it is crucial to note that these results are largely computational and based on currently available sequence data. Limitations exist in terms of the number and diversity of samples analyzed, and the conclusions drawn are probabilistic rather than definitive.

## Figures and Tables

**Figure 1 animals-13-02617-f001:**
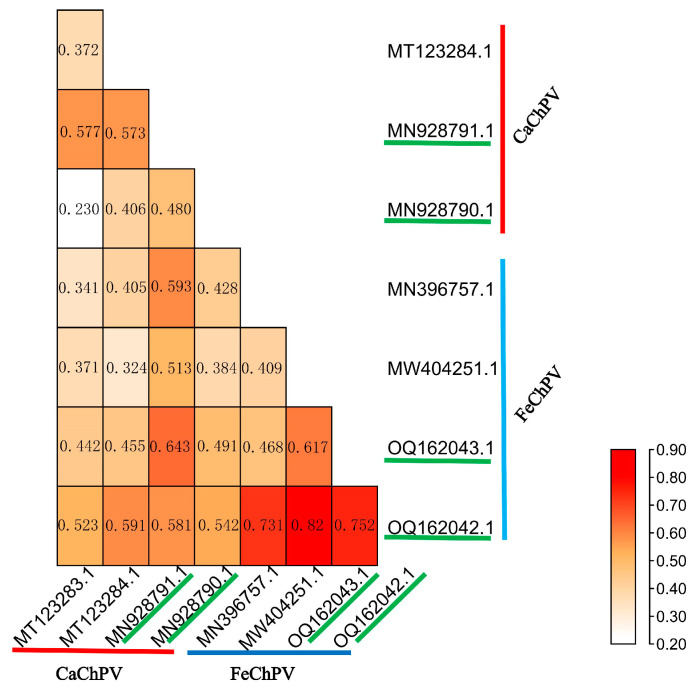
Heat map of root-mean-square deviation of FeChPV and CaChPV. MT123283.1, MY123284.1, MN928791.1, and MN928790.1 refer to CaChPV; MW396757.1, MW404251.1, OQ162043.1, and OQ162.42.1 refer to FeChPV. Among them, MN928791.1, MN928790.1, OQ162043.1, and OQ162042.1 are cross-species transmitted strains.

**Figure 2 animals-13-02617-f002:**
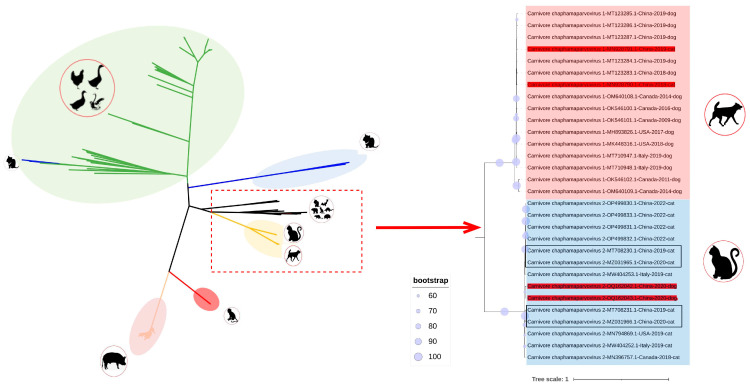
Phylogenetic trees of *Chaphamaparvovirus* based on complete sequence (*n* = 351). Potential cross-species transmission strains are highlighted in red. The feline chaphamaparvovirus obtained in this study is highlighted in rectangles. The best substitution model was GTR + F + R10. The outgroup *Ursus americanus* chapparvovirus/MN166196.1 branch not shown here.

**Figure 3 animals-13-02617-f003:**
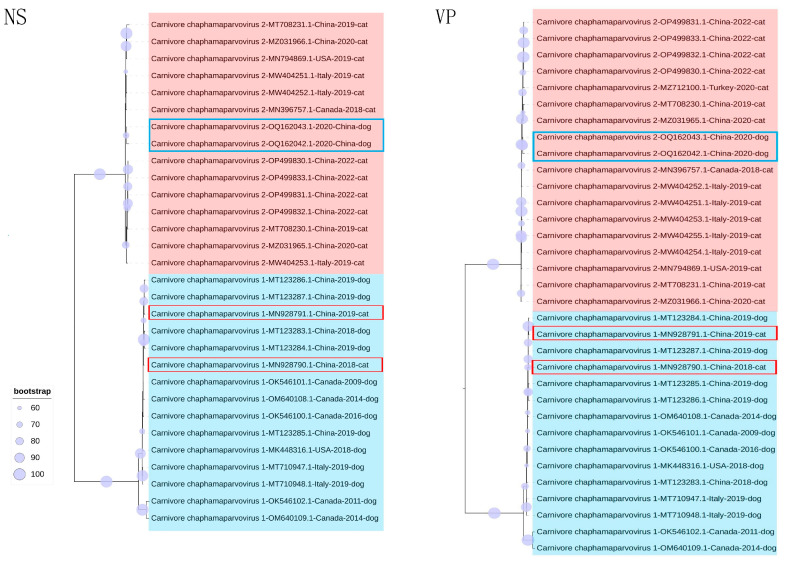
Phylogenetic trees of feline chaphamaparvovirus (FeChPV) and canine chaphamaparvovirus (CaChPV) based on NS and VP. NS (*n* = 30); VP (*n* = 33). The best substitution model of NS was TPM2u + F + G4, and that of VP was TPM3 + F + G4. The *Ursus americanus* chapparvovirus/MN166196.1 was used as an outgroup (not shown). A red background indicates FeChPV; a blue background indicates CaChPV; Potential cross-species transmission strains are highlighted in red. Interactive Tree of Life was used for visualization. NS, nonstructural protein; VP, virion protein.

**Figure 4 animals-13-02617-f004:**
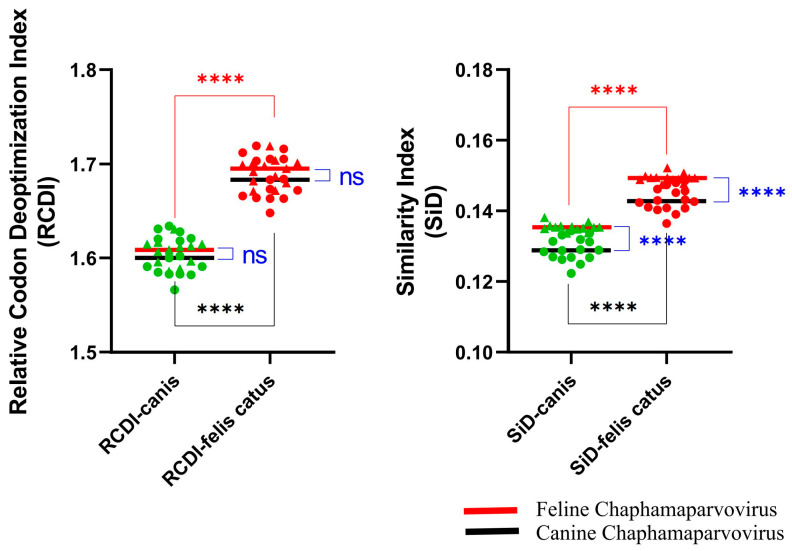
Codon usage pattern difference between FeChPV/CaChPV and hosts. RCDI: relative synonymous codon usage analysis; SiD: similarity index analysis. The black line represents the mean of the RCDI or SiD of CaChPV, and the red line represents the mean of the D*p* of FeChPV. Each dot (triangle or circle) represents a strain, triangles represent FeChPV, circles represent CaChPV. Green dots represent dogs as target host, while red dots represent cats as target host. *Canis* represents *Canis lupus familiaris*. Parametric and non-parametric *t*-tests were used to analyze the significance of Gaussian distribution data and non-conforming data, respectively. **** *p* < 0.0001; ns, *p* > 0.05.

**Figure 5 animals-13-02617-f005:**
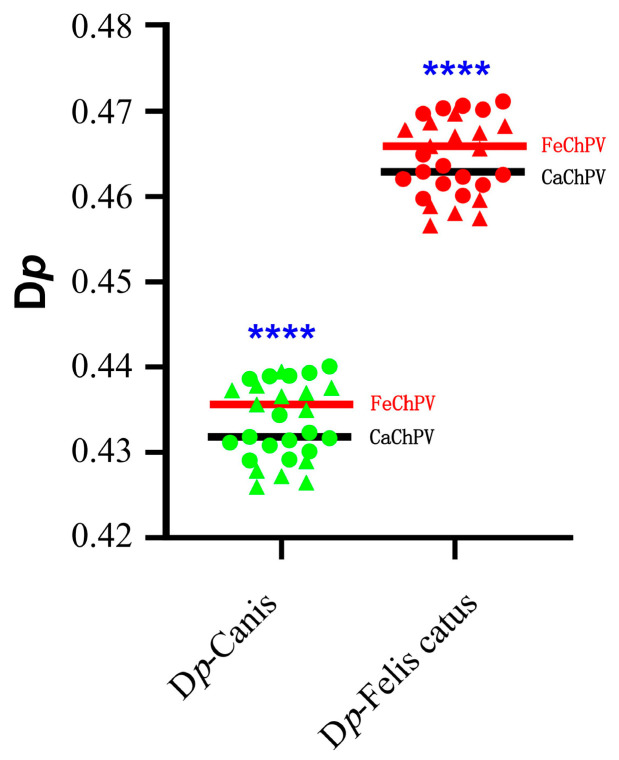
Analysis of synonymous codon usage preferences between viruses and hosts. The black line represents the mean of the D*p* of CaChPV, and the red line represents the mean of the D*p* of FeChPV. Each dot (triangle or circle) represents a strain, triangles represent FeChPV, circles represent CaChPV. Green dots represent dogs as target host, while red dots represent cats as target host. *Canis* represents *Canis lupus familiaris*. The Shapiro–Wilk test was performed for the normality test. Parametric and non-parametric t tests were used to analyze the significance of Gaussian distribution data and non-conforming data, respectively. **** *p* < 0.0001; *p* > 0.05 is unlabeled.

**Figure 6 animals-13-02617-f006:**
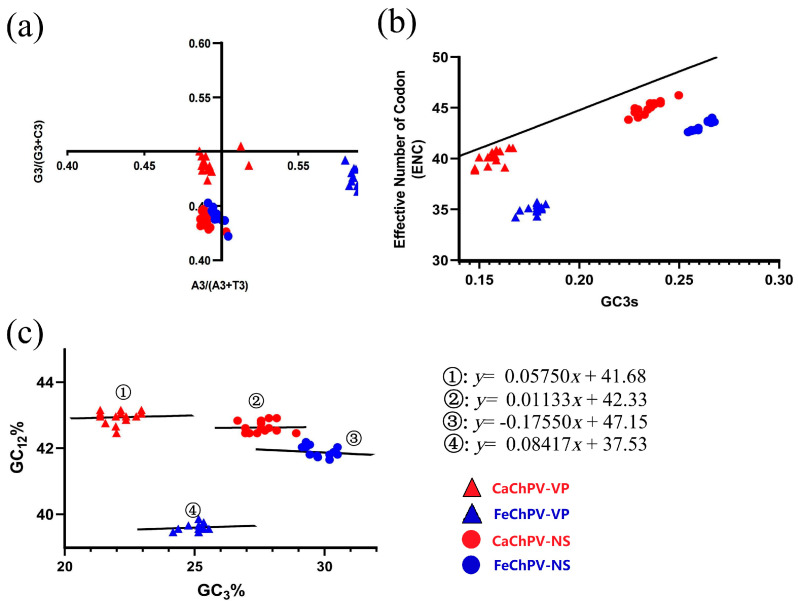
Parity rule 2, ENC plot, and neutrality analysis based on NS and VP of FeChPV and CaChPV. (**a**) Parity rule 2; (**b**) ENC plot; (**c**) neutrality analysis. Red circles and triangles represent the NS and VP of CaChPV; blue circles and triangles represent the NS and VP of FeChPV. NS, nonstructural protein; VP, virion protein; ENC, effective number of codons.

**Table 1 animals-13-02617-t001:** The codon adaptation index of feline chaphamaparvovirus (FeChPV) and canine chaphamaparvovirus (CaChPV).

	Viruses	Dogs (*Canis lupus* Familiaris) (Mean ± SD)	Cats (*Felis catus*) (Mean ± SD)
CAI	FeChPV	**0.7624 ± 0.0008**	0.7089 ± 0.0009
	CaChPV	**0.7485 ± 0.0020**	0.6950 ± 0.0020
eCAI	FeChPV	**0.7694 ± 0.0017**	0.7157 ± 0.0016
	CaChPV	**0.7657 ± 0.0029**	0.7113 ± 0.0030
Normalised CAI (CAI/eCAI)	FeChPV	**0.9909 ± 0.0024**	0.9906 ± 0.0033
	CaChPV	**0.9775 ± 0.0038**	0.9771 ± 0.0042

Note: The larger values between FeChPV and CaChPV are highlighted in bold.

**Table 2 animals-13-02617-t002:** The effective number of codons of feline chaphamaparvovirus (FeChPV) and canine chaphamaparvovirus (CaChPV).

Virus	Gene	Range of ENC	The Average of ENC (X ± S)
FeChPV	*NS*	42.6–44.0	43.123 ± 0.461
	*VP*	34.2–35.7	35.081 ± 0.417
CaChPV	*NS*	43.9–46.3	45.047 ± 0.613
	*VP*	39.1–41.2	40.16 ± 0.761

**Table 3 animals-13-02617-t003:** Mutation analysis of cross-species transmission FeChPV and CaChPV strains.

Strains	Mutation Sites
CaChPV-MN928790.1	I56T, Y68C, F131S, D402N, **R449K**
CaChPV-MN928791.1	Y22H, Q365R, **R449K**, H484P,
FeChPV-OQ162042.1	M19T, **G112R**, F151S, **L174P**, **S197T**, S206A, **S325R**, E352G, S445N,
FeChPV-OQ162043.1	**G112R, L174P, **S197T**, S325R,**

Note: Mutations that occurred in both strains are bolded. Mutations to the same site as the other viral species are underlined.

**Table 4 animals-13-02617-t004:** B-cell immune epitopes of FeChPV and CaChPV strains.

Strains	Location	Epitope	Score
CaChPV-MN928790.1	388–407	WGPWTWKDIYGIGSNTRMYS	1.000
	475–494	PEMIEMQELHHTDDEEIEVI	0.980
	123–142	WKDSSMKDSSIAYKEGMFKS	0.908
	24–43	NNTLATIVAAETGGNAINTG	0.797
	363–382	TTQGCFQVTLHLACKKRRSR	0.758
CaChPV-MN928791.1	479–498	EMQELPHTDDEEIEIITADE	1.000
	388–407	WGPWTWKDIYGIGSDTRMYS	0.858
	24–43	NNTLATIVAAETGGNAINTG	0.724
Other CaChPV strains	475–494	PEMIEMQELHHTDDEEIEVI	/ *
	388–407	WGPWTWKDIYGIGSDTRMYS	/
	24–43	NNTLATIVAAETGGNAINTG	/
	123–142	WKDSSMKDSSIAYKEGMFKS	/
	363–382	TTQGCFQVTLHLACKKRRSR	/
FeChPV-OQ162042.1	144–163	VTNPLKDSSIAYKEGMFKQG	1.000
FeChPV-OQ162043.1	145–164	TNPLKDFSIAYKEGMFKQGT	1.000
Other FeChPV strains	145–164	TNPLKDFSIAYKEGMFKQGT	1.000

Note: * Scores were different between strains. Identical amino-acid sequences are underlined.

## Data Availability

The data that support the findings of this study are available from the Appendix A.

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
