# Peer review of "Phylogenetic Analysis and Codon Usage Bias Reveal the Base of Feline and Canine Chaphamaparvovirus for Cross-Species Transmission"

_animals, 2023, doi:10.3390/ani13162617_

Round 1
Reviewer 1 Report (New Reviewer)
It is a good manuscript describing association of CaChPV and FeChPV and possible transmission from one species to other through various softwares and analysis tools.
There are certain suggestions:
1. Some genus have been written without species; such as Canis and Ursus. However towards the end of the manuscript authors have indicated species of Ursus. I feel uniformly genus and species of all should be mentioned.
2. Some of the sentences needs correction and I have marked them on the manuscript with editing too.
3. In the figure 2 Feline Chaphamaparvovirus is mentioned where as Carnivore Chapparvovirus is mentioned which needs correction as both are Carnivore Chamhamaparvovirus 1 and 2 respectively.
4. In the results sequencing of genome (from commercial vendor or done at in-house facility) is missing

I am not a very good expert to comment on the quality of English. However, I felt that the manuscript need some fine trimming.
Author Response
Dear Reviewer and Editor,
Thank you for your valuable feedback on our manuscript. We have carefully considered your suggestions and made the necessary revisions. Here are our responses to your specific points:
Reviewer1
It is a good manuscript describing association of CaChPV and FeChPV and possible transmission from one species to other through various softwares and analysis tools.
There are certain suggestions:
- Some genus have been written without species; such as Canis and Ursus. However towards the end of the manuscript authors have indicated species of Ursus. I feel uniformly genus and species of all should be mentioned.
Response: Regarding the inclusion of species names for all genera mentioned in the manuscript, we appreciate your observation. We have now uniformly mentioned the genus and species for all relevant taxa. Additionally, the image section has not been modified for aesthetic purposes, but an explanation has been added to the caption.
- Some of the sentences needs correction and I have marked them on the manuscript with editing too.
Response: Thank you for your correction. We acknowledge the need for correction in certain sentences, as indicated by your editing marks on the manuscript. We have carefully reviewed these sentences and made the necessary revisions according to your suggestions as shown in
Line 43-44, 46, 242-243, 380-381.
- In the figure 2 Feline Chaphamaparvovirus is mentioned where as Carnivore Chapparvovirus is mentioned which needs correction as both are Carnivore Chamhamaparvovirus 1 and 2 respectively.
Response: We sincerely thank you for bringing the discrepancy in Figure 2 to our attention. We have addressed this issue by redrawing the figure, ensuring that both Carnivore Chapparvovirus strains are correctly labeled as Carnivore Chamhamaparvovirus 1 and 2 respectively for better readability and comprehension.
- In the results sequencing of genome (from commercial vendor or done at in-house facility) is missing
Response: We appreciate your comment regarding the missing information about genome sequencing in the results section, specifying that it was performed by a commercial vendor as shown in Line 198-200.

Reviewer 2 Report (New Reviewer)
Guo et al. present a detailed overview of what is known about the sequence composition of members of the genus Chaphamaparvovirus. Their analyses are thorough, but additional clarification is needed to better link the results to the conclusions.
Major remarks:
- I think the article would really benefit from a direct comparison of the different CaChPV and FeChPV strains on the nucleotide and amino acid level in the beginning of the manuscript. In its current state, the article does not clearly indicate how divergent these two viruses are and how this compares to the divergence between strains of the same virus. This is important information to be able to better understand the results of the analyses performed in the rest of the manuscript.
- On a related note, the impact of the sequence similarity between the different virus strains on the different analyses should be discussed. Given the small size of these virus’ genomes, minor, spurious changes could potentially have a big impact on the observed values but this is difficult to assess without a detailed understanding of the sequence composition of these viruses. For example, the RSCU value for a certain codon could be significantly altered with 1/a few mutations if the overall occurrence of that specific codon is low but codon counts are not provided, making this difficult to assess.
- Some of the analyses were performed not only for cats and dogs but also for other vertebrate hosts. However, these latter results are not included in the manuscript and only provided as raw data in the supplementary files. However, these values seem comparable to those of cats and/or dogs, which seems to undermine some of the conclusions. Please include these results also in the figures/tables of the main manuscript and discuss their relevance for the conclusions being drawn.
Minor comments:
Line 17: ‘’For members of the genus Chaphamaparvovirus”.
Line 24: Names of viruses are never written in italics. As this sentence refers to all members of the genus Chaphamaparvovirus, the correct way to phrase this would be: “Chaphamaparvoviruses are ancient viruses that have been …”.
Line 45: “and its members can infect …”. A genus is an abstract concept, it does not infect anything. Infection is done by the viruses classified within a certain taxon.
Line 47: “tasmanian devils”, no italics.
Line 47: “humans”.
Line 48: “are classified within the species …”
Line 52: “cachavirus”, no capital.
Line 57: “than that in”
Line 84: “and”, no italics.
Line 120: Are you sure “concatenated” is the correct term here?
Line 139: Higher or lower? In the results section you mention that the lower values of CaChPV seem to indicate better host adaptation.
Line 205: “and”, no italics.
Line 216-219: Figure 1: is the tree on the right a close-up of a section of the tree on the left? Or is it a separately made tree using the same sequences? You state that ‘Potential cross-species transmission strains are highlighted in red’ but there are red circles in this figure, a red dashed line, a red arrow, a red section of the tree… It is unclear to which strains you are referring. Please also include a metric for branching support (eg bootstrap values) on the tree.
Line 221: Figure 2: Please include a metric for branching support (eg bootstrap values) on the tree.
Line 229: This should be paragraph 3.3
Line 230: Which codon usage pattern difference? This is the first time you mention this…
Line 232-233: “are better adapted”
Line 232-233: According to figure 3 the difference is not significant, can you then really claim this?
Line 232-233: Looking at Supplementary Table 2, other hosts seem to have better values than cats and even dogs for CAI, RSID and SiD. Please include this data in Figure 3 and Table 1 and discuss the relevance of these findings.
Line 235: Here you state yourself that the difference is not significant but you still conclude that CaChPV is more adapted. This conclusion is not supported by your analysis.
Line 248: “than that to”
Line 254-255: “This finding also strongly suggests that FeChPV demonstrates a higher degree of adaptability to dogs and cats. These results are consistent with the results for RCDI and SiD”. I fail to see what you base this conclusion on and how these results are consistent with the RCDI/SiD analyses.
Line 257-258: Is this because there are insufficient sequences available or because you didn’t include enough of them in your analysis? Some metrics here would be useful. If the problem is statistical errors, then why don’t you correct these errors?
Line 258-259: How exactly does the combination of these metrics suggest the possibility of host spillover?
Line 262: Nothing is highlighted in bold.
Line 316-317: Do you mean that position 131 in CaChPV corresponds to position 151 in FeChPV? Or do you just mean that both genomes have an F->S mutation somewhere? In what way is this a relevant finding?
Line 318: Table 3: You list the mutation sites of these four sequences, but which reference are you using? To which sequence are these ones being compared?
Line 318: Table 3: Are these the only differences between these strains? Does this mean that almost all differences between FeChPV and CaChPV are silent mutations? I think it would be interesting to include an extra figure/table in the beginning of the manuscript to indicate the intra- and inter-virus diversity of CaChPV and FeChPV on both the nucleotide and amino acid level. Interpreting the codon usage analyses data would be lot easier if the reader had a clear perception of the overall divergence of FeChPV and CaChPV.
Line 379: Is there really a statistically significant difference between the values for cats and dogs? The values for dogs are slightly higher, but can you truly rule out the role of cats based on this difference?
The quality of the English language is good, a few minor linguistic and typographic errors need to be corrected (see above).
Author Response
Dear Reviewer and Editor,
Thank you for your valuable feedback on our manuscript. We have carefully considered your suggestions and made the necessary revisions. Here are our responses to your specific points:
Reviewer 2
Guo et al. present a detailed overview of what is known about the sequence composition of members of the genus Chaphamaparvovirus. Their analyses are thorough, but additional clarification is needed to better link the results to the conclusions.
Major remarks:
- I think the article would really benefit from a direct comparison of the different CaChPV and FeChPV strains on the nucleotide and amino acid level in the beginning of the manuscript. In its current state, the article does not clearly indicate how divergent these two viruses are and how this compares to the divergence between strains of the same virus. This is important information to be able to better understand the results of the analyses performed in the rest of the manuscript.
Response: Thank you for your suggestion. We agree with your suggestion to include a direct comparison of the nucleotide and amino acid sequences of different CaChPV and FeChPV strains in the beginning of the manuscript. It will provide a clear understanding of the level of divergence between these two viruses and how it compares to the divergence between strains of the same virus. In the revised manuscript, we have included a comprehensive comparison of these sequences, enabling readers to better comprehend the results of our subsequent analyses. Please refer to lines 204-212 and 368-379.
- On a related note, the impact of the sequence similarity between the different virus strains on the different analyses should be discussed. Given the small size of these virus’ genomes, minor, spurious changes could potentially have a big impact on the observed values but this is difficult to assess without a detailed understanding of the sequence composition of these viruses. For example, the RSCU value for a certain codon could be significantly altered with 1/a few mutations if the overall occurrence of that specific codon is low but codon counts are not provided, making this difficult to assess.
Response: Thank you for your suggestion. We fully recognize the impact of sequence similarity on the various analyses performed and its potential influence on observed values. We agree that minor changes in the genome of these small-sized viruses could have a significant impact on results such as codon usage and relative synonymous codon usage (RSCU). To address this concern, we have added a discussion on the impact of sequence similarity between virus strains in the manuscript. Additionally, we have provided the counts of codon usage in the supplementary table 3, allowing readers to assess the potential impact of spurious changes on the observed values.
- Some of the analyses were performed not only for cats and dogs but also for other vertebrate hosts. However, these latter results are not included in the manuscript and only provided as raw data in the supplementary files. However, these values seem comparable to those of cats and/or dogs, which seems to undermine some of the conclusions. Please include these results also in the figures/tables of the main manuscript and discuss their relevance for the conclusions being drawn.
Response: Thank you for your comments. We have added relevant content to the discussion. During the generation of the phylogenetic tree, we discovered a close relation of FeChpV and CaCHpV to these animals. As such we decided to analyze these data and discovered similar characteristics to those of canines and cats. We do not believe this weakens our conclusions. As CaChPV was detected in cats and FeChPV was detected in dogs, we analyzed the CAI, RCDI, Sid, and other features to further elucidate the possibility of cross-species transmission However, FeChpv and CaChPV have not been detected in other animal species and it is uncertain whether superinfection with both viruses is possible. Therefore, the analysis for other animals strongly suggests that further investigations and studies should be conducted to explore whether there are more frequent cross-species transmission events of ChPV. In addition, given that ChPV is an ancient virus and that FeChPV and CaChPV show a high degree of adaptation to a marsupial in our data, it is unlikely that the transmission journey was from primitive marsupials to other more modern domestic animals. In conclusion, we believe that the analysis for other animals is relevant, but at first we mistakenly thought that this section seemed irrelevant to the topic of this paper and therefore we removed it in last revision. After rethinking, and based on your suggestion, we think it is necessary to mention the results in the article. Please refer to Lines 423-432. Thank you for your constructive comments.
Minor comments:
Line 17: ‘’For members of the genus Chaphamaparvovirus”.
Response: Thank you for your correction. We have revised the sentence accordingly as shown in Line 17.
Line 24: Names of viruses are never written in italics. As this sentence refers to all members of the genus Chaphamaparvovirus, the correct way to phrase this would be: “Chaphamaparvoviruses are ancient viruses that have been …”.
Response: We appreciate your correction. The sentence has been updated as suggested as shown in Line 24.
Line 45: “and its members can infect …”. A genus is an abstract concept, it does not infect anything. Infection is done by the viruses classified within a certain taxon.
Response: Thank you for your correction, we have revised the sentence accordingly as shown in Line 45.
Line 47: “tasmanian devils”, no italics.
Response: We apologize for the incorrect formatting. The italics have been removed as shown in Line 46.
Line 47: “humans”.
Response: Thank you for catching that omission. We have added the missing "s" as shown in Line 47.
Line 48: “are classified within the species …”
Response: Thank you for your correction. We have revised the sentence as suggested. Please refer to Line 48.
Line 52: “cachavirus”, no capital.
Response: We appreciate your correction. The sentence has been revised accordingly as shown in Line 52.
Line 57: “than that in”
Response: Thank you for your correction. We have made the necessary revision as shown in 57.
Line 84: “and”, no italics.
Response: We apologize for the formatting error. The italics have been removed as shown in Line 84.
Line 120: Are you sure “concatenated” is the correct term here?
Response: Thank you for your comment. We have reevaluated the term and made the necessary revision. Please refer to Line 121.
Line 139: Higher or lower? In the results section you mention that the lower values of CaChPV seem to indicate better host adaptation.
Response: Thank you for your comment. We have modified this sentence to make it clearer. Please refer to Lines 140-141.
Line 205: “and”, no italics.
Response: We appreciate your correction. The sentence has been revised accordingly as shown in 216.
Line 216-219: Figure 1: is the tree on the right a close-up of a section of the tree on the left? Or is it a separately made tree using the same sequences? You state that ‘Potential cross-species transmission strains are highlighted in red’ but there are red circles in this figure, a red dashed line, a red arrow, a red section of the tree… It is unclear to which strains you are referring. Please also include a metric for branching support (eg bootstrap values) on the tree.
Response: Thank you for your question. Figure 1 has been redrawn. The right part of Figure 1 shows the tree made using the same sequence. Moreover, we have added bootstrap values to improve the interpretation of the tree.
Line 221: Figure 2: Please include a metric for branching support (eg bootstrap values) on the tree.
Response: Thank you for your comments. We have redone Figure 2, standardized the virus strain nomenclature, and added bootstrap values for better interpretation.
Line 229: This should be paragraph 3.3
Response: We apologize for the mistake. The error has been corrected, and the paragraph is now correctly labeled as 3.3.
Line 230: Which codon usage pattern difference? This is the first time you mention this…
Response: Thank you for your comment. We have rewritten this sentence to make it better understandable as shown in Lines 243-244.
Line 232-233: “are better adapted”
Response: We appreciate your comment. The phrase has been revised to provide a more accurate description as shown in Line 246.
Line 232-233: According to figure 3 the difference is not significant, can you then really claim this?
Response: Thank you for pointing out the inconsistency. We have revised the sentence and made the necessary adjustments in Figure 3 to accurately represent the significance of the differences. In lines 232-233 of the original manuscript, this sentence means that in the difference between the two viruses against different hosts, the difference is significant, and we have modified Figure 3 to express it more clearly. Non-significant difference means that the difference in RCDI values between FeChPV and CaChPV against the same host is not significant. To avoid confusion for the reader, we have modified this part of the expression (Lines 243-248) and Figure 3.
Line 232-233: Looking at Supplementary Table 2, other hosts seem to have better values than cats and even dogs for CAI, RSID and SiD. Please include this data in Figure 3 and Table 1 and discuss the relevance of these findings.
Response: Thank you for your comment. At the beginning of our analysis, we also analyzed hosts with strains closely related to CaChPV and FeChPV, such as Sarcophilus harrisii and Mus musculus. However, we considered that this paper was mainly focused on dogs and cats and analyzing other animals seemed to be a noisy exercise. The adaptation analysis for other animals we found high adaptation of FeChPV and CaChPV to certain animals, especially Sarcophilus harrisii, which strongly suggests a strong potential for cross-host transmission of FeChPV and CaChPV. Whether ChPV will undergo frequent host spillover events like astravirus is worth further exploration and a question. Although FeChPV and CaChPV show high fitness for some animals, no evidence of FeChPV or CaChPV infection has been found in other animals, so we cannot associate high fitness with evidence of infection. We believe that your suggestion is correct and therefore we have added a related discussion in the Discussion section. Please refer to Lines 423-432.
Line 235: Here you state yourself that the difference is not significant but you still conclude that CaChPV is more adapted. This conclusion is not supported by your analysis.
Response: Thank you for your comments. The section and Figure 3 have been revised. The insignificance of the difference is defined as the difference in the same host where FeChPV and CaChPV are not significant for the same host, while in FeChPV and CaChPV are significant for different hosts. We believe that this result can support the conclusion that " CaChPV and FeChPV were more adapted to dogs than to cats". Please refer to Lines 243-248.
Line 248: “than that to”
Response: Thank you for your correction. It has been modified as shown in Line 262.
Line 254-255: “This finding also strongly suggests that FeChPV demonstrates a higher degree of adaptability to dogs and cats. These results are consistent with the results for RCDI and SiD”. I fail to see what you base this conclusion on and how these results are consistent with the RCDI/SiD analyses.
Response: Thank you for your comments. This section has been overhauled and, as you said, we found that the sentence did not make sense, so we have revised and removed parts of it.
Line 257-258: Is this because there are insufficient sequences available or because you didn’t include enough of them in your analysis? Some metrics here would be useful. If the problem is statistical errors, then why don’t you correct these errors?
Response: Thank you for pointing out this ridiculous mistake. We found this one result inconsistent with the RCDI and Sid results, and we think it is due to too few available sequences even though we downloaded almost all available sequences from NCBI. It is because of this disagreement that we do not discuss in our conclusions which one of FeChPV or CaChPV is more adapted to a particular host, so you we think that this difference does not affect our conclusions. We believe that there is no human statistical error and that we use three metrics, CAI, eCAI and nCAI, to further correct for the bias introduced by a single metric. This part has been modified as shown in Line 267.
Line 258-259: How exactly does the combination of these metrics suggest the possibility of host spillover?
Response: Thank you for your correction. It may be too arbitrary to present this conclusion here, so we have removed this sentence.
Line 262: Nothing is highlighted in bold.
Response: Thank you for your correction, we have made the relevant corrections. Please refer to Table 1.
Line 316-317: Do you mean that position 131 in CaChPV corresponds to position 151 in FeChPV? Or do you just mean that both genomes have an F->S mutation somewhere? In what way is this a relevant finding?
Response: Thank you for your question. Regarding the section on mutations, we are focusing on four potentially cross-species-transmissible strains. Given that in feline and canine parvovirus, site mutations can cause affinity of the vp protein to the transferrin receptor and thus cross-species transmission can occur. Therefore, we focused on point mutations in these four strains. The S to F mutation at position 151 of the vp protein in FeChPV is also present in CacHPV. Position 151 of FeChPV is located at the corresponding position 131 of CaChPV.
Line 318: Table 3: You list the mutation sites of these four sequences, but which reference are you using? To which sequence are these ones being compared?
Response: Thank you for your question. The four strains we listed are potentially cross-species-transmissible strains. We have compared these four strains with all VP protein sequences to analyze their mutational characteristics, please refer to our supplementary table 6.
Line 318: Table 3: Are these the only differences between these strains? Does this mean that almost all differences between FeChPV and CaChPV are silent mutations? I think it would be interesting to include an extra figure/table in the beginning of the manuscript to indicate the intra- and inter-virus diversity of CaChPV and FeChPV on both the nucleotide and amino acid level. Interpreting the codon usage analyses data would be lot easier if the reader had a clear perception of the overall divergence of FeChPV and CaChPV.
Response: Thank you for your suggestions. These four strains are potential recombinant sequences. Considering that point mutations may cause changes in affinity with the host receptor, we focused on analyzing the point mutation characteristics of these four strains, but the ones listed in Table III are not the only differences. Mutations common to the mutant strains, or mutations in which CaChPV is identical to FeChPV, are listed. Mutations in other strains are not listed. We have uploaded a map of mutations that clearly reflects the mutation characteristics among all strains and among the four listed strains. Please refer to the supplementary table 6.
Line 379: Is there really a statistically significant difference between the values for cats and dogs? The values for dogs are slightly higher, but can you truly rule out the role of cats based on this difference?
Response: Thank you for your correction. This sentence was incorrect and we have revised it. In Figure 3, we found that the difference for FeChPV and CaChPV for the same host does not seem to be significant, so this sentence is incorrect, thank you again for your correction, we have revised this sentence as shown in Lines 401-432.

Reviewer 3 Report (New Reviewer)
The paper "Phylogenetic Analysis and Codon Usage Bias Reveal the Potential of Feline and Canine Chaphamaparvovirus for Cross-species Transmission" is a well designed and written manuscript on a newly discovered virus on pet animals. I have only one comment that I think needs to be addressed before acceptance: why don't you search for other disease in differential diagnosis for gastroenteritis? Please add some sentences to explain this and how that can affect the interpretation of the results, regarding the pathogenicity of the virus.
Author Response
Dear Reviewer and Editor,
Thank you for your valuable feedback on our manuscript. We have carefully considered your suggestions and made the necessary revisions. Here are our responses to your specific points:
Reviewer 3:
The paper "Phylogenetic Analysis and Codon Usage Bias Reveal the Potential of Feline and Canine Chaphamaparvovirus for Cross-species Transmission" is a well designed and written manuscript on a newly discovered virus on pet animals. I have only one comment that I think needs to be addressed before acceptance: why don't you search for other disease in differential diagnosis for gastroenteritis? Please add some sentences to explain this and how that can affect the interpretation of the results, regarding the pathogenicity of the virus.
Response: Thank you for your comments. We found that ChPV has an extremely broad host range and more importantly, we found that ChPV demonstrates cross-species transmission potential in dogs and cats. Evolution and research have shown that the virus has the ability to spread across species between dogs and cats. This provides a strong clinical management imperative: healthy cats should not be in close contact with dogs with CPV. We therefore conducted an in-depth study for ChPV to explore its potential for cross-species transmission. However, this virus has been less studied in terms of its pathogenicity, and some studies have detected ChPV in dogs and cats with respiratory and gastrointestinal symptoms. Further studies into ChPV pathogenicity in dogs and cats are warranted and should not be limited to FeChPV or CaChPV.

Round 2
Reviewer 2 Report (New Reviewer)
Thank you for your corrections. I have only a few remaining remarks:
Line 24: There is more than a single chaphamaparvovirus. As such, this statement should be written in plural: Chaphamaparvoviruses are …
Line 44: This sentence is still incorrect. Chaphamaparvovirus is a taxon name that has no relation to the biological entities classified within. As such, it cannot infect anything. Furthermore, taxon names are never abbreviated. Please correct this to: “… subfamily, viruses within the genus Chaphamaparvovirus can infect …”. The abbreviation ChPV can be introduced on the following sentence.
Line 238: Virus names are never written (in part) in italics (see the ICTV FAQs: “A virus name should never be italicized, even when it includes the name of a host species or genus, and should be written in lower case. This ensures that it is distinguishable from a species name, which otherwise might be identical”). Please correct this to Ursus americanus chapparvovirus. The same correction should also be made on line 238.
Line 263: Something seems to have gone wrong with this correction. It should be “than to cats”.
Line 370: Seeing that similarity and genetic distance are inversely related, I assume you mean that the within-group similarity is higher and the genetic distance lower compared to between groups.
Author Response
Dear Reviewer and Editor,
Thank you for your valuable feedback on our manuscript quickly. We have carefully considered your suggestions and made the necessary revisions. Here are our responses to your specific points:
Comments and Suggestions for Authors
Thank you for your corrections. I have only a few remaining remarks:
Line 24: There is more than a single chaphamaparvovirus. As such, this statement should be written in plural: Chaphamaparvoviruses are …
Respond: Thanks for the correction again, we have added "es" as shown in line 24
Line 44: This sentence is still incorrect. Chaphamaparvovirus is a taxon name that has no relation to the biological entities classified within. As such, it cannot infect anything. Furthermore, taxon names are never abbreviated. Please correct this to: “… subfamily, viruses within the genus Chaphamaparvovirus can infect …”. The abbreviation ChPV can be introduced on the following sentence.
Respond:We appreciate your further correction and we have revised this sentence as shown in Line 44.
Line 238: Virus names are never written (in part) in italics (see the ICTV FAQs: “A virus name should never be italicized, even when it includes the name of a host species or genus, and should be written in lower case. This ensures that it is distinguishable from a species name, which otherwise might be identical”). Please correct this to Ursus americanus chapparvovirus. The same correction should also be made on line 238.
Respond:Thanks to your careful correction, we have removed the italics as shown in lines 232 and 238.
Line 263: Something seems to have gone wrong with this correction. It should be “than to cats”.
Respond:Thanks to your careful correction, we have removed the “that” as shown in lines 262.
Line 370: Seeing that similarity and genetic distance are inversely related, I assume you mean that the within-group similarity is higher and the genetic distance lower compared to between groups.
Respond: We appreciate your correction, we are aware of the error and have fixed the sentence, please refer to lines 369 to 373.
This manuscript is a resubmission of an earlier submission. The following is a list of the peer review reports and author responses from that submission.
Round 1
Reviewer 1 Report
In this manuscript Xu et al. presents the detection of feline chaphamaparvoviruses and the bioinformatical analyzes of the codon usage and composition of this virus. The in silico examinations were also carried out in case of canine chaphamaparvovirus. Overall, the manuscript is difficult to understand and follow; it has plenty of inaccurate wording choices, which make it hard for the reader to comprehend if given information refers to the host, the virus, or both. The taxonomy is poorly presented, with the authors constantly referring to the Parvovirinae, which is a subfamily and not the name of the family. Furthermore, Chaphamaparvovirus is a genus, not a species, of a distinct subfamily from the Parvovirinae. The biggest issue, however, is the concept of the manuscript, as parvoviruses (and many other small ssDNA virus families) have been shown previously to exhibit little adaptation to their host as far as codon usage is concerned, hence very little would this contribute to the ability of host switching. As the manuscript depicts a similar scenario to what has already happened in the Protoparvovirus genus, it would have been essential to execute the same analysis in case of these viruses, i.e., canine parvovirus, feline panleukopenia virus and mink enteritis virus. Furthermore, the manuscript keeps referring to chaphamaparvoviruses as ancient, yet never mentions or considers for the analysis the possible arthropod origin of this virus clade. There is evidence that these viruses are of an "ancient" origin in arthropods but none whatsoever that this is true for vertebrates as well.
Specific points
Overall: all official names of virus taxonomy should be in Italics. Latin names in zoology are always in Italics if referring to a species or genus. Animal families and any higher level taxa should not be written in Italics.
Line 27: which species of pangolin, bat, rodent, hedgehog or primate?
Line 40: Parvoviridae is the family, not a subfamily
Line 42: Chaphamaparvovirus is a genus, not a species.
Line 50: What is meant here? Within genus Chaphamaparvovirus the VP protein sequence is actually more conserved than the NS. If this refers to the whole family, only one domain of the NS is actually conserved throughout.
Line 79: the exact species of the animal should be named here
Line 85: What was the substitution model?
Line 152: What is meant here? Virus was detected in 58 samples? Later on only four samples are being discussed as positive.
Line 158: Does this refer to genes, ORFs or genomes?
Figure 1 is very difficult to read.
Line 186: reference needed. Which viruses? This statement is inaccurate if the whole virusphere is considered.
Line 197: what does this sentence mean?
Table 2: the number of same high frequency codons appears to vary on quite a large scale to me.
Lines 203 to 225: How are these CAI values related to any given gene of any random organism compared to these hosts? I am not convinced that these CAI values are indeed due to adaptation and not just the consequence of parvoviruses in general to prefer AT-rich codons, the number of which is limited, hence it will cause a CAI bias. The difference between the canine and feline host is minimal and not sure it could be considered significant.
Line 145 and throughout the paragraph: selection pressure to what?
Line 255: same taxonomy inaccuracies as already mentioned.
Line 257: Reference needed. Moreover, this statement is rather generic: could be true for any parvovirus.
Line 271: In what way are these two viruses closely related to these animals?
Ethical concern: vertebrate animals were subjects of these experiment yet there is no ethics statement under which these examinations were permitted to be carried out.
Author Response
Rewiew1
In this manuscript Xu et al. presents the detection of feline chaphamaparvoviruses and the bioinformatical analyzes of the codon usage and composition of this virus. The in silico examinations were also carried out in case of canine chaphamaparvovirus. Overall, the manuscript is difficult to understand and follow; it has plenty of inaccurate wording choices, which make it hard for the reader to comprehend if given information refers to the host, the virus, or both. The taxonomy is poorly presented, with the authors constantly referring to the Parvovirinae, which is a subfamily and not the name of the family. Furthermore, Chaphamaparvovirus is a genus, not a species, of a distinct subfamily from the Parvovirinae. The biggest issue, however, is the concept of the manuscript, as parvoviruses (and many other small ssDNA virus families) have been shown previously to exhibit little adaptation to their host as far as codon usage is concerned, hence very little would this contribute to the ability of host switching. As the manuscript depicts a similar scenario to what has already happened in the Protoparvovirus genus, it would have been essential to execute the same analysis in case of these viruses, i.e., canine parvovirus, feline panleukopenia virus and mink enteritis virus. Furthermore, the manuscript keeps referring to chaphamaparvoviruses as ancient, yet never mentions or considers for the analysis the possible arthropod origin of this virus clade. There is evidence that these viruses are of an "ancient" origin in arthropods but none whatsoever that this is true for vertebrates as well.
Specific points
Overall: all official names of virus taxonomy should be in Italics. Latin names in zoology are always in Italics if referring to a species or genus. Animal families and any higher level taxa should not be written in Italics.
Respond: Thank you for your correction. We have checked it from beginning to end and corrected the wrong parts, thank you again for your help.
Line 27: which species of pangolin, bat, rodent, hedgehog or primate?
Respond: Due to the word limit we did not write the corresponding species in detail. In Materials and methods we added detailed species information as shown in Line 78-82.
Line 40: Parvoviridae is the family, not a subfamily
Respond: Thank you for your correction, we have revised this sentence as shown in Line 40-41.
Line 42: Chaphamaparvovirus is a genus, not a species.
Respond: Thank you for your correction, we have revised this sentence as shown in Line 42.
Line 50: What is meant here? Within genus Chaphamaparvovirus the VP protein sequence is actually more conserved than the NS. If this refers to the whole family, only one domain of the NS is actually conserved throughout.
Respond: Thank you for your correction, we have deleted this sentence to avoid ambiguity.
Line 79: the exact species of the animal should be named here
Respond: Thank you for your suggestion. We have added the information about the species as shown in Line 78-82. There are some exact species in GenBank that we could not trace (such as bat, pangolin, while some are of multiple subspecies origin and therefore not labeled with their exact Latin species names (such as porcine, poultry/birds).
Line 85: What was the substitution model?
Respond: The substitution model, also known as the evolutionary model or molecular substitution model, is a mathematical representation of the process by which genetic mutations occur over time. These models are essential for constructing phylogenetic trees, which are graphical representations of the evolutionary relationships among various species or other taxa. In the context of phylogenetic tree construction, the substitution model is used to estimate the number of nucleotide or amino acid substitutions that have occurred between different sequences. This estimation is used to infer the evolutionary distance between the sequences, which helps build the tree and understand the relationships among the taxa.
Line 152: What is meant here? Virus was detected in 58 samples? Later on only four samples are being discussed as positive.
Respond: Yes, as you said, 4 positive samples were detected out of 58 samples. To avoid ambiguity, we have modified the expression of this sentence as shown in Line 160-161.
Line 158: Does this refer to genes, ORFs or genomes?
Respond: The original sentence was wrong, thank you for your correction, we have fixed this sentence as shown in Line 167-168.
Figure 1 is very difficult to read.
Respond: Thanks to your suggestion, Figure 1 has been redrawn and divided into two figures.
Line 186: reference needed. Which viruses? This statement is inaccurate if the whole virusphere is considered.
Respond: Thanks for your comment. Thank you very much for the correction. We have revised the formulation of the sentence and added the reference as shown in Line 204-205.
Line 197: what does this sentence mean?
Respond: This sentence may be disturbing to the reader, so we have deleted it. And, in Discussion, we have modified the relevant expression to make it easier to understand as shown in lines 318-322.
Table 2: the number of same high frequency codons appears to vary on quite a large scale to me.
Respond: Thanks for your comment. As you said, there is a large gap in the use of the same high frequency codons. It reflects to some extent the variability of codon usage patterns between FeChPV/CaChPV and the host.
Lines 203 to 225: How are these CAI values related to any given gene of any random organism compared to these hosts? I am not convinced that these CAI values are indeed due to adaptation and not just the consequence of parvoviruses in general to prefer AT-rich codons, the number of which is limited, hence it will cause a CAI bias. The difference between the canine and feline host is minimal and not sure it could be considered significant.
Respond: Thank you for your comment. The viral relationship to other organisms and genes remains unclear, as this study has focused exclusively on the viral relationship to the specific types of host. The phylogenetic tree has shown that FeChPV and CaChPV are closely related to ChPV; thus, we examined the risk of cross-species transmission. The CAI value is only one of the relevant reference factors. The CAI value has been used to assess the expression of an exogenous gene within the cell. A high CAI value represents high levels of gene expression. When a virus infects a cell, the virus, being an exogenous gene, uses the host material, such as tRNA, to express its own genes. Over time, some viruses gradually adapt to the host to increase the expression levels of their genes, which affects their chances of survival and propagation. We used CAI values to assess differences in adaptation against different hosts, which is a measure of expression ability within a host cell. Among-host differences in codon use lead to different CAI values of viruses in different hosts. Therefore, we believe that the preference for codons with A/T endings and CAI values may be reconcilable. We have added relevant explanations to the discussion as shown in Line 314-316 and 341-344.
Line 145 and throughout the paragraph: selection pressure to what?
Respond: Thank you for your question. I think you are talking about line 245. Selection pressure in virus evolution refers to the various factors that drive changes in viral populations over time. These pressures can influence the survival and replication of viral strains, with some variants becoming more successful due to their specific adaptations. The selection pressure drives the process of natural selection, which can result in the emergence of new viral strains with improved fitness in a given environment. We have added relevant explanations to the discussion as shown in Line 350-356.
Line 255: same taxonomy inaccuracies as already mentioned.
Respond: Thank you for your correction, we have revised this sentence as shown in lines 281.
Line 257: Reference needed. Moreover, this statement is rather generic: could be true for any parvovirus.
Respond: Thank you for your correction. We have added references in the appropriate places as shown in lines 283. Also, as you said, this statement is generic, but here we are mainly emphasizing that Chaphamaparvovirus is also an old virus.
Line 271: In what way are these two viruses closely related to these animals?
Respond: Thank you for your correction. We have revised the sentence as shown in lines 301-303.
Ethical concern: vertebrate animals were subjects of these experiment yet there is no ethics statement under which these examinations were permitted to be carried out.
Respond: Thanks to your comments, we have provided ethical approval information like the editor and added relevant information in the manuscript as shown in lines 74-76.
Reviewer 2 Report
The present study aimed to investigate the potential for Chaphamaparvovirus to cross the species barrier by means of phylogenetic analysis and codon usage analysis.
This study is interesting and I consider that it may be suitable for publication in the journal Animal.
Just one observation, the sequence with accession number MT708230 in the GenBank is annotated as Dependovirus genus and not as Chaphamaparvovirus; please review the sequence and correct the annotation in the GenBank or make the correction in the article.
Is there information about which cells type FeChPV and/or CaChPV infect? What receptors does the viral ligand recognize? It would be important to incorporate information in this regard that reinforces or rejects the hypothesis about cross-host transmission. Why were cats with respiratory infections not included in the study? Improve the quality of the phylogenetic tree of figure 1a, since it is not possible to observe the association between the viral genomesPlease explain why the phylogenetic tree that includes sequences from the NS region and that groups FeChPV and CaChPV suggesting the possibility of cross-species transmission events, however, the phylogenetic tree that includes sequences from the VP region that does not group the sequences of ChPV of felines and dogs and that is the principal region that is related to the viral infection to the cell, does not show these cross-species transmission events
Author Response
Review 2
The present study aimed to investigate the potential for Chaphamaparvovirus to cross the species barrier by means of phylogenetic analysis and codon usage analysis.
This study is interesting and I consider that it may be suitable for publication in the journal Animal.
Just one observation, the sequence with accession number MT708230 in the GenBank is annotated as Dependovirus genus and not as Chaphamaparvovirus; please review the sequence and correct the annotation in the GenBank or make the correction in the article.
Respond:Thank you for your comments. This strain (MT708230) was isolated from our laboratory. In fact, it is a Chaphamaparvovirus, not a dependoparvovirus, and there were some errors in the uploading process to NCBI. We have contacted Genbank and will revise the annotation in GenBank, which may take a while. Thank you again for your corrections.
Is there information about which cells type FeChPV and/or CaChPV infect? What receptors does the viral ligand recognize? It would be important to incorporate information in this regard that reinforces or rejects the hypothesis about cross-host transmission.
Respond: Thank you for your comments. There are fewer studies on Chaphamaparvovirus, especially in dogs and cats. Information on the types of cells and receptors it infects is unclear. At first, we tried to analyze the ability of the VP protein of the virus to bind to the host receptor, but were not sure what its receptor was for entry into the cell and therefore gave up. We therefore analyze its potential for cross-species transmission only from the perspective of codon usage.
Why were cats with respiratory infections not included in the study?
Respond: The chaphamaparvovirus is considered a diarrhea-associated virus; therefore, during sample collection, we focused only on diarrhea symptoms. However, it was also detected in respiratory diseases, which we did not consider. However, respiratory disease analysis was outside the study scope, as we focused on the host overall, while using swab samples from the respiratory tract; consequently, the present findings remain valid. we do not think that this affects the results of this study, because we targeted the host as a whole, rather than analyzing viral effects on a particular organ. We have added relevant explanations to the discussion as shown in Line 290-294.
Improve the quality of the phylogenetic tree of figure 1a, since it is not possible to observe the association between the viral genomes
Respond: Thanks to your suggestion, Figure1 has been refactored
Please explain why the phylogenetic tree that includes sequences from the NS region and that groups FeChPV and CaChPV suggesting the possibility of cross-species transmission events, however, the phylogenetic tree that includes sequences from the VP region that does not group the sequences of ChPV of felines and dogs and that is the principal region that is related to the viral infection to the cell, does not show these cross-species transmission events
Respond: Thank you for your question. We redrew Figure 1, which shows more clearly the relationship between canine and feline Chaphamaparvovirus. As shown in Figure 1, two strains of Chaphamaparvovirus isolated from cats clustered with canine Chaphamaparvovirus in both the NS and VP, indicating a potential cross-species transmission event.
Reviewer 3 Report
From the outset, I feel this a very molecular virological study that would be better placed in a journal specialising in these matters (with more specialist and appropriate reviewers) rather than one with the much broader title of “Animals”. I am neither a molecular virologist nor a parvovirus expert so cannot give the fine analysis required for the M&M. I do not understand the work on codons and how the authors are able to make predictions about what other mammalian species these canine and feline viruses might most readily infect. This is my ignorance of course and I leave to other reviewers to advise this journal. Therefore, I make only some general observations and then one or two specifics
This submission makes the case for two closely related parvoviruses, of domestic cat and of domestic dogs, being potentially able to cross infect from one species to the other, and also addresses the additional possibility that these viruses may infect other species such as man who associates very closely with these domestic species. In current times such possible spill-overs and the outcomes these may have are topical, and this work goes to lengths to show the potential. In my opinion, the possibility of any viruses changeing ship is a foregone conclusion but I appreciate the work that has gone into trying to understand some of the molecular processes that facilitate these cross transferences. Having said this, however, I feel that the discussion could be restructured and simplified to make things easier for the broader readership.
Some specific points that require clarification are:
Lines 40/41 Is the last word of the first sentence correct?
Lines 271/272. Viruses are not related to mammals – they may be related to the viruses of these named mammals.
Lines 275/275. Much of the previous text in this paragraph concerns the genetic relationships between viruses and mammals. Then we get the following sentence “Humans and primates are closely related to dogs and cats as companion animals”. This reads as though there is some genetic link between people and companion animals. It also suggests that monkeys may be keeping pets which I doubt they are, not yet anyway. Perhaps this sentence could be rewritten as “Humans are closely associated with dogs and cats as companion animals” ?
Lines 298/299: Has this not been said already in introduction or elsewhere?
Author Response
Review 3
From the outset, I feel this a very molecular virological study that would be better placed in a journal specialising in these matters (with more specialist and appropriate reviewers) rather than one with the much broader title of “Animals”. I am neither a molecular virologist nor a parvovirus expert so cannot give the fine analysis required for the M&M. I do not understand the work on codons and how the authors are able to make predictions about what other mammalian species these canine and feline viruses might most readily infect. This is my ignorance of course and I leave to other reviewers to advise this journal. Therefore, I make only some general observations and then one or two specifics
This submission makes the case for two closely related parvoviruses, of domestic cat and of domestic dogs, being potentially able to cross infect from one species to the other, and also addresses the additional possibility that these viruses may infect other species such as man who associates very closely with these domestic species. In current times such possible spill-overs and the outcomes these may have are topical, and this work goes to lengths to show the potential. In my opinion, the possibility of any viruses changeing ship is a foregone conclusion but I appreciate the work that has gone into trying to understand some of the molecular processes that facilitate these cross transferences. Having said this, however, I feel that the discussion could be restructured and simplified to make things easier for the broader readership.
Some specific points that require clarification are:
Lines 40/41 Is the last word of the first sentence correct?
Respond: Thank you for your correction. We have revised the sentence as shown in lines 40-41.
Lines 271/272. Viruses are not related to mammals – they may be related to the viruses of these named mammals.
Respond: Thank you for your correction. We have revised the sentence as shown in lines 301-303.
Lines 275/275. Much of the previous text in this paragraph concerns the genetic relationships between viruses and mammals. Then we get the following sentence “Humans and primates are closely related to dogs and cats as companion animals”. This reads as though there is some genetic link between people and companion animals. It also suggests that monkeys may be keeping pets which I doubt they are, not yet anyway. Perhaps this sentence could be rewritten as “Humans are closely associated with dogs and cats as companion animals” ?
Respond: Thank you for your careful reading and corrections. As you said, the original sentence had the word "primates" in it, and we have changed this sentence as shown in lines 304-305.
Lines 298/299: Has this not been said already in introduction or elsewhere?
Respond: Thank you for your comment, we have deleted the sentence to avoid unnecessary duplication
Round 2
Reviewer 1 Report
Although the authors did revise the manuscript and addressed some of the specific points raised, the major flaws still persist. This is the concept of the study; as I mentioned in my previous report, there have been several similar studies executed on parvoviruses and none of them are referenced in the paper or the findings being compared. The very event, namely the host switch between cats and dogs, has already happened in the Protoparvovirus genus, yet this is never analyzed or compared to the findings of the study. The statistics, such as the CAI, still lack proper controls. The responses given to my comments did not seem to address the actual question at hand, which still persist regarding the statistics. Because of these issues, unfortunately, my recommendation remains unchanged.